# Evidence of Orientation-Dependent Early States of Prion Protein Misfolded Structures from Single Molecule Force Spectroscopy

**DOI:** 10.3390/biology11091358

**Published:** 2022-09-16

**Authors:** Andrea Raspadori, Valentina Vignali, Anna Murello, Gabriele Giachin, Bruno Samorì, Motomasa Tanaka, Carlos Bustamante, Giampaolo Zuccheri, Giuseppe Legname

**Affiliations:** 1Department of Neuroscience, Scuola Internazionale Superiore di Studi Avanzati (SISSA), 34136 Trieste, Italy; 2Dipartimento di Farmacia e Biotecnologie, Alma Mater Studiorum Università di Bologna, 40126 Bologna, Italy; 3Department of Chemical Sciences (DiSC), University of Padua, 35131 Padova, Italy; 4Laboratory for Protein Conformation Diseases, RIKEN Brain Science Institute, Wako 351-0198, Saitama, Japan; 5QB3 Institute, University of California, 642 Stanley Hall #3220, Berkeley, CA 94720-3220, USA; 6S3 Center of the Institute of Nanoscience of the Italian National Research Council (CNR), 41125 Modena, Italy; 7Interdepartmental Center for Industrial Research on Health Science and Technologies, University of Bologna, 40126 Bologna, Italy; 8ELETTRA Sincrotrone Trieste S.C.p.A, Basovizza, 34139 Trieste, Italy

**Keywords:** prions, atomic force microscopy, single molecule force spectroscopy, protein misfolding, intrinsically disordered proteins

## Abstract

**Simple Summary:**

Prion diseases are neurodegenerative disorders caused by the amyloidal aggregation of the cellular prion protein. We apply single-molecule force spectroscopy approaches to study the unfolding of prion protein monomers and dimers in different orientations. We find heterogeneous behavior in the prion protein unfolding and an interesting difference between the dimer orientations whereby the dimer in which the C-termini are joined unfolds at a higher force, implying a more stable structure owing to interactions between the C-termini. These results may contribute to a better understanding of the initial steps of oligomer assembly during prion diseases.

**Abstract:**

Prion diseases are neurodegenerative disorders characterized by the presence of oligomers and amyloid fibrils. These are the result of protein aggregation processes of the cellular prion protein (PrP^C^) into amyloidal forms denoted as prions or PrP^Sc^. We employed atomic force microscopy (AFM) for single molecule pulling (single molecule force spectroscopy, SMFS) experiments on the recombinant truncated murine prion protein (PrP) domain to characterize its conformations and potential initial oligomerization processes. Our AFM-SMFS results point to a complex scenario of structural heterogeneity of PrP at the monomeric and dimer level, like other amyloid proteins involved in similar pathologies. By applying this technique, we revealed that the PrP C-terminal domain unfolds in a two-state process. We used two dimeric constructs with different PrP reciprocal orientations: one construct with two sequential PrP in the N- to C-terminal orientation (N-C dimer) and a second one in the C- to C-terminal orientation (C-C dimer). The analysis revealed that the different behavior in terms of unfolding force, whereby the dimer placed C-C dimer unfolds at a higher force compared to the N-C orientation. We propose that the C-C dimer orientation may represent a building block of amyloid fibril formation.

## 1. Introduction

The propagation of misfolded proteins is a hallmark of a wide variety of diseases, including neurodegenerative disorders, such as prion diseases (or transmissible spongiform encephalopathies, TSE), Alzheimer’s disease (AD), Parkinson’s syndrome and amyotrophic lateral sclerosis [1]. Despite the proteins involved in these diseases being different, the mechanisms of protein aggregation have common features: protein monomers assemble into oligomers, which in turn form protofibrils and finally large amyloid fibers [2]. Protein oligomers have acquired increasing importance as they are supposed to be the cell’s main toxic species [3,4]. However, the structural characterization of oligomeric states is extremely difficult as multiple states can coexist. Single-molecule (SM) techniques are well suited to exploring such complex and heterogeneous folding landscapes as they can characterize rare and transient states. Single-molecule force spectroscopy (SMFS) has been used to reconstruct folding pathways and to observe transient structures, enabling a quantitative description of these processes [5]. Conformational equilibria of intrinsically disordered proteins have been extensively characterized using SMFS performed through coupled atomic force microscopy (AFM-SMFS) [6], optical tweezers (OT) force spectroscopy [7] and SM fluorescence [8].

Specifically, in AFM-SMFS, the AFM probe is used to stretch a macromolecule (e.g., a protein) which is tethered between the probe and the substrate surface. The deflection of the calibrated AFM cantilever measures the applied force (through Hooke’s law), while the extension is measured. An AFM instrument for performing SMFS consists of a piezoelectric positioner used to stretch the macromolecule. The AFM probe, the macromolecule and the substrate are all immersed in a buffer solution. Commonly, surface adsorbed proteins are attached to the AFM probe by pushing the probe into the protein layer with a controlled force: such attachment can withstand the pulling forces necessary to unfold commonly folded proteins [9,10].

AFM-SMFS approaches are uniquely suitable for tackling the challenge with transient features of amyloid and intrinsically disordered proteins causing neurodegenerative diseases, such as, for instance, the amyloid-β peptide (the peptide involved in AD) or α-synuclein, a protein implicated in Parkinson’s syndrome. The mechanical unfolding of the monomeric prion protein, i.e., the causal agent of TSE, has been previously explored by means of OT-SMFS experiments. Few studies employing experimental approaches similar to AFM-SMFS are available, and they mainly use monomeric or fibrillar forms of the recombinant prion protein.

TSE are the prototypical misfolding diseases and include Creutzfeldt–Jakob disease in humans, scrapie in sheep and goats, and chronic wasting disease in cervids. The structural conversion of the α-helical folded cellular prion protein (PrP^C^) into its pathological form, PrP^Sc^, causes TSE [11]. The very recently solved cryo-electron microscopy (EM) near-atomic structures of infectious, brain-derived PrP^Sc^ fibrils unveiled a continuum of β-strand serpentine threading of the protein C-terminal domain and following studies have confirmed that the PrP^Sc^ amyloids feature parallel in-register β-strand stack folding [12,13,14,15,16,17]. However, the conformational conversion of PrP^C^ into PrP^Sc^ appears as a multi-step process whose molecular mechanisms still remain unclear and are extremely difficult to probe at the single-molecule level.

In an effort to gain new insights into the initial phases of protein aggregation processes, we applied SMFS to the characterization of the unfolding events occurring in both monomeric and dimeric forms of the recombinant α-helical folded truncated mouse prion protein (hereafter denoted as PrP, from residue 89 to 230) when mechanically pulled.

To stretch PrP molecules by AFM-SMFS approaches, a series of PrP constructs were designed that included monomeric and dimeric PrP forms, each flanked by 4 GB1 protein modules on both N- and C-termini (Figure 1).

GB1 was chosen as the reference protein module as it has the shortest unfolding length with respect to other available and mechanically characterized proteins that have been used for such a purpose: it has an unfolding or contour length, ΔLc, of about 18 nm. The lengths of the protein chain segments that unfold in each consecutive mechanically-induced unfolding event are reflected by ΔLc, and GB1 polyprotein constructs have characteristic unfolding patterns, well-defined ΔLc and unfolding forces that can serve as internal control modules to validate single-molecule interactions [18,19]. It is customary to flank the protein of interest by two series of GB1 domains, such as 4 on each side, in order to obtain easily recognizable saw-tooth pattern force-extension curves. It is also a way to be sure that the protein of interest has been stretched whenever it is possible to recognize and count a number of GB1 unfolding events that would require the stretching of the macromolecule section, including the protein of interest (such as from 5 to 8 GB1 events, when 4 GB1′s flank on both sides).

The dimeric PrP form has been previously reported as an important intermediate in oligomerization [20]. Therefore, the investigation of mechanical unfolding of dimeric PrP constructs may shed light on the very early events of prion conversion and assembly, which are still unknown.

Our AFM-SMFS measurements confirmed a significant conformational diversity in PrP, involving contacts along the entire polypeptide chain. Interestingly, dimeric PrP unfolding forces appear higher when two C-terminal domains are placed in close proximity.

## 2. Materials and Methods

### 2.1. Production and Purification of Hetero-Polymeric Constructs

Plasmids of protein constructs were designed and purchased from GenScript (Piscataway, NJ, USA). Constructs contained either 8 GB1 modules alone, denoted as (GB1)_4_–(GB1)_4_, or monomeric or dimeric murine PrP (UniProt entry P04925 from residues 89 to 230) forms, each flanked by 4 GB1 protein modules on both N- and C-termini.

For clarity, the PrP-containing constructs are referred to as (a) (GB1)_4_PrP(GB1)_4_, which refers to a monomeric PrP flanked by GB1 modules; (b) (GB1)_4_PrP_2_(GB1)_4_ N-C, which refers to PrP dimers where the N-terminus of the second PrP molecule is linked to the C-terminus of the first molecule and separated by a short Arg-Ser linker sequence; and (c) (GB1)_4_PrP_2_(GB1)_4_ C-C, where the C-terminus of the second PrP molecule is linked to the C-terminus of the first molecule (Figure 1A). The C-C constructs were derived from two individual PrP containing 4 GB1 at the N-terminus and an extra Cys after codon 230 (also denoted as the Ins231C mutant). The linkage was obtained by means of a Cys231-Cys231230230 disulfide bond oxidized in vitro (see below).

The proteins were expressed in *E. coli* BL21 (DE3) cells at 30 °C. Natively and soluble-expressed soluble proteins were purified by FPLC using HisTrap and size exclusion chromatography (SEC) approaches. The supernatant from lysed cells was loaded slowly at 0.5 mL/min onto a 5 mL HisTrap crude FF (GE Healthcare) column mounted on Äkta Purifier system (GE Healthcare) previously equilibrated with binding buffer (25 mM Trizma-Base, 150 mM NaCl, pH 8.0). The column was washed with washing buffer (25 mM Trizma-Base, 150 mM NaCl, pH 8.0) to remove non-specifically bound proteins, and then elution was performed by means of a linear gradient of elution buffer (25 mM Trizma-Base, 150 mM NaCl, 500 mM Imidazole, pH 8.0). The purification of the Ins231C mutant was performed in the presence of 1 mM 1,4-dithiothreitol (DTT) in both buffers to prevent protein concatenation and to maintain Cys231 in a reduced state. The purified protein was buffer exchanged in the same buffer without DTT and subsequently allowed to spontaneously dimerize in an oxidizing environment at + 4 °C for 48 h. A second purification step for all constructs was performed using a SEC Sephacryl S-300 10/60 (GE Healthcare) column equilibrated in washing buffer (25 mM Trizma-Base, 150 mM NaCl, pH 8.0).

The protein purity was assessed by SDS-PAGE gel (Appendix A) and the protein folding was also verified by circular dichroism (CD) showing two characteristic minima at around 208 and 222 nm that confirm the presence of both α-helical and β-sheet secondary structures in the folding of the protein constructs used in this study (Appendix A). All CD spectra have been collected on a JASCO model J-810 spectropolarimeter coupled with a Peltier system. The CD measurements were recorded by averaging two scans in the 190–240 nm range, using a bandwidth of 2 nm and a time constant of 1 s at a scan speed of 50 nm/min. Spectra were acquired at a protein concentration of around 2–4 μM, using HELLMA quartz cells with Suprasil windows and an optical path length of 0.1 cm.

### 2.2. Force Spectroscopy Experiments and Data Analysis

Constant-velocity mechanical unfolding experiments were performed with a Bruker Picoforce AFM on a Multimode Nanoscope IIIa controller (Bruker). Doubly gold-coated, V-shaped silicon nitride cantilevers with a nominal spring constant of 0.06 N/m were used (NPG, Bruker). Unfolding experiments were performed on the homomeric polyprotein (GB1)_4_–(GB1)_4_, (GB1)_4_PrP(GB1)_4_, and dimeric PrP constructs, i.e., (GB1)_4_PrP_2_(GB1)_4_ N-C and (GB1)_4_PrP_2_(GB1)_4_ C-C in 20 mM Tris HCl, pH 7.4 buffer. All the experiments have been performed at a temperature of approximately 28 °C in the AFM closed fluid cell (as measured with a thermocouple in our room-temperature equilibrated fluid cell). A drop of protein constructs (10 μL at 4–10 μM) was deposited on a flame-cleaned glass coverslip and adsorbed for 20 min. The fluid cell was then filled with neutral pH buffer. Thermal tuning was performed in the respective solution to determine the cantilever spring constant. The pulling speed was 2180 nm/s. Data filtering and peak fitting were performed using custom-developed software [21] and previously published protocols [6]. Briefly, curve peaks were fitted using the worm like chain (WLC) model following the equation below [22]:(1)$$F=\frac{k_{b}T}{p}\;\left[\frac{x}{L}+\frac{1}{4}\;{\left(1-\frac{x}{L}\right)}^{2}-\frac{1}{4}\right]$$
where *k_b_* is the Boltzmann constant, *T* is temperature, *L* is contour length and *p* is the persistence length. In our study, we set *p* = 0.4 nm. We considered 1 nm spatial sensitivity as the result of an overestimation by a factor of two, the sum of the experimental error and WLC fitting. Analysis of force and ΔLc distributions was performed using custom-made software in Matlab (Mathworks, Natick, MA, USA). 

## 3. Results

The monomeric and dimeric PrP constructs were flanked by 4 GB1 protein modules on both N- and C-termini. To mimic the tandem modular design of natural elastomeric proteins such as titin, we used a polyprotein GB1 construct consisting of eight identical tandem repeats of GB1 domains. Previous studies demonstrated that GB1 polyprotein exhibits a combination of mechanical features, including fast, high-fidelity folding kinetics, low mechanical fatigue and ability to fold against residual force that make GB1 particularly ideal for AFM-SMFS approaches [18,23]. Furthermore, with its 18 nm average unfolding length, GB1 is the smallest protein module in use as a SMFS internal standard, so it was chosen in order to allow a facile determination of the predicted long unfolding events due to protein aggregation. Dimeric PrP were obtained either by an engineered construct encoding for two tandem-arranged PrP segments oriented from the N- to C-termini (hereafter N-C) or by a disulfide linkage between two PrP with an added C-terminal Cys, achieving a C- to C-terminal orientation (C-C) (Figure 1A). The addition of GB1 modules linked to PrP allowed the expression of soluble polyproteins that were then purified by SEC prior to each SMFS experiment to remove aggregated forms (Appendix A). The force-unfolding events of the monomeric PrP construct were compared with the control made of the 8 GB1 modules (Figure 1B,C). As we and others already reported [18,23], stretching 8 GB1 results in force–extension curves (FEC) of characteristic saw-tooth pattern appearance (Figure 1B), with an average unfolding force of about 200  pN for GB1 domains, which is similar to the mechanical stability of the I27 domain from the natural elastomeric protein titin [24].

In the FEC of Figure 1B,C, each peak, except the detachment peak, has an unfolding force (denoted as F) and a contour-length increment (ΔLc) which can be extracted for the analysis. Calculating the difference between the L values of two separated rupture events, the change of delta contour length (ΔLc) can be then obtained. This parameter gives the exact length of an unfolded protein module [25]. A FEC for the mechanical unfolding of (GB1)_4_PrP(GB1)_4_ was considered valid if it had at least five unfolding peaks with ΔLc of 18.5 nm ± 3 nm (i.e., characteristic of GB1) so that the included PrP module had been certainly stretched. The unfolding peaks in the curve of Figure 1B,C can be interpreted as GB1 modules (up to eight in total) or as the unfolding of PrP.

While the GB1 octamer control displays only very few unfolding events occurring at contour lengths (ΔLc) longer than 18 nm, the inclusion of a PrP monomer in the construct brings a significant occurrence of events at unfolding lengths in the ΔLc 23–40 nm range (i.e., longer than the GB1 distribution). These unfolding events are not tightly clustered around any specific contour length or unfolding force, but rather they are scattered (Figure 2 and Appendix A). The maximum theoretical unfolding length of a complete PrP fragment in AFM-SMFS is about 21 nm. It thus resulted that the monomeric truncated PrP showed a number of rare unfolding events spanning the unfolding length range between 23 and 40 nm, with an apparently continuous distribution in unfolding forces. This behavior witnesses a conformational heterogeneity of PrP in a significant fraction of the molecules. When molecules are not folded natively in the construct, they can create mechanically stable non-native contacts that we record as long-unfolding events.

A quantitative description of the distribution of the unfolding events with a ΔLc in the range of GB1 unfolding (around ΔLc = 18 nm in the *inset* of Figure 2) led us to a further characterization of the molecular system.

Differential analysis of the mechanical unfolding of constructs containing GB1 with or without monomeric PrP exhibits a marked and significant difference. As evident from the kernel density estimation (KDE) “heat maps” in Figure 3A (see also Appendix A for the individual maps), a clear increase in the number of events at about 19–24 nm unfolding length and at ~200 pN unfolding force is recorded. Even though a partial overlap with the GB1 unfolding distribution is present, these events are clearly distinguishable and they are numerically equivalent to about 0.8 ± 0.2 events per poly-protein unfolding event. Due to the unfolding length values and the numerical consistency, we can interpret that this unfolding region mainly pertains to the native monomeric PrP structure. This unfolding contour length is in very good agreement with the known C-terminal structured portion of PrP that is expected to measure about 21 nm when fully stretched [26]. We can conclude that PrP has a native fold in most of the protein constructs we adsorbed on the surface and that AFM SMFS unfolding experiments can characterize the conformational heterogeneity of PrP from single molecule data.

As a control, we verified our interpretation of this region in the distribution of unfolding events by preparing a poly-protein construct including only the C-terminal portion of the truncated PrP protein (residues 125–230), flanked by the four GB1 modules at both N- and C-termini. With this, we recorded a force-distance distribution of events that was practically indistinguishable from that of the full PrP construct in the region at less than 23 nm unfolding length, with the same numeric consistency of native PrP events in the range of 19–24 nm unfolding length (Figure 3B). Such a construct lacking the N-terminal portion of the truncated PrP showed no events in the 24–40 nm region, as expected by the lack of the N-terminal section and confirming that these are due to interactions requiring the full length of the truncated PrP chain.

As additional proof of the unfolding of the PrP native structures, we performed force spectroscopy after chemical reduction of the intramolecular disulfide bridge of PrP. Such a reduction has been reported to dramatically alter the stability of native PrP folding [27]. Our measurements of the mechanical unfolding of constructs containing the reduced PrP module show no additional stable structures in the 19–23 nm region (Figure 3C). A slightly more narrowly peaked distribution for GB1 has been observed, possibly due to a lower experimental error in determining the initial point of chain stretching, together with an increase in the unfolded chain length in agreement with a fully unfolded PrP polypeptide chain (Appendix A). The constructs with the chemically reduced PrP monomer display a slightly lower frequency of long unfolding events at 24–40 nm.

To gain insight into the early aggregation processes, two different dimeric PrP constructs were prepared with flanking GB1: (GB1)_4_PrP_2_(GB1)_4_, where the two PrP moieties display different known reciprocal orientations. In the first one, the C-terminal of the first PrP moiety was linked to the N-terminal of the second PrP moiety, resulting in an N- to C-terminal orientation, i.e., (GB1)_4_PrP_2_(GB1)_4_ N-C construct. In the second, the two C-termini of (GB1)_4_PrP were linked together in a C-C orientation via a disulfide bridge between two identical protein molecules, using a C-terminal cysteine appended at position 231, i.e., the (GB1)_4_PrP_2_(GB1)_4_ C-C construct. This preparation strategy leads to unequivocal orientation of PrP in the constructs and enables the study of the effect of the vicinity and of the orientation on the emergence of new associative structures and on the stability of the C-terminal folded portion. Constructs containing PrP dimers in both N-C or C-C orientations display a comparable number of events not only in the 24–40 nm range of non-native unfolding contour-lengths but also in the even longer 40–80 nm range of unfolding lengths (Appendix A). As these long unfolding lengths are never present in the monomeric PrP construct, we interpreted these unfolding events as characteristic unfoldings of the PrP dimeric constructs. Unfolding lengths of up to 80 nm would comprise the full lengths of the two PrP chains and they could occur only if the native folding of both PrP was missing. The number and the distribution of unfolding lengths and unfolding forces for the 24–40 nm and the 40–80 nm unfoldings do not appear statistically different in the C-C and N-C dimeric constructs (Appendix A). Unfoldings at lengths larger than 80 nm are virtually absent, confirming the lack of signals due to nonspecific interactions between PrP and GB1.

The analysis of the results of force spectroscopy on the constructs containing dimeric PrP showed a markedly different behavior in the region of the natively folded PrP structure as a function of the reciprocal dimer orientation. While the N-C dimers showed unfolding signals analogous (see Figure 4A versus Figure 3A) to the natively folded structure found in monomeric PrP, the C-C dimers lacked such signals completely (Figure 4A,B).

By comparing the unfolding signals of the two constructs comprising dimeric PrP, we noticed that the C-C dimers contained a new highly populated region of unfolding events at higher force and at a shorter unfolding length than the native PrP folding: an unfolding force of 250–330 pN and 17–21 nm, i.e., even higher force than the unfolding of GB1 (Figure 4C), which is found at the usual lower force range (e.g., centered at about 210 pN). It turned out that about 0.5 events per poly-protein unfolding were found in this new region, as compared to 1.1 events found in the native PrP region in the N-C dimer, i.e., still lower than twice the 0.8 events per unfolding in the monomer (Appendix A). The estimates of the size of these populations could be affected by some experimental error, as these regions overlap with the more numerous GB1 module unfolding events. Nonetheless, the almost complete disappearance of the unfoldings of the native PrP structures and the emergence of this new structure with half the occurrence strongly suggested that this might represent the emergence of a new and more stable associative structure that could involve parts of the two neighboring C-terminal structures. In the dimers, the structured C-terminal portions are covalently bound, thus their effective relative concentration is significantly higher than in a solution of monomers, possibly destabilizing their native structure to some extent and facilitating their association.

The linking of the two folded structures through the unfolded N-terminal section leads to them “seeing” each other with an augmented concentration with respect to being independent in solution [28]. Using the WLC model [22], we can estimate a reciprocal concentration of about 10–20 mM for the C-terminal portions linked in an N-C dimer. Such a high concentration might be the reason for a lower occurrence of folded native structures than was expected (1.1 versus 1.6 per polyprotein pulled). When the C-terminal sections are instead directly linked through their C termini in the C-C dimers, their relative concentration is much higher, possibly reaching the molar range. It is thus not surprising that their interaction could take place and the formation of associative structures could be very fast. Quite likely, the vicinity of the two natively folded domains in the C-C dimers could pose some hindrance to their correct folding or, alternatively, lead to the stabilization of folding due to interactions between folded structures. The different geometry of the constructs could imply different degrees of molecular crowding, an effect that is expected to lead to the stabilization of folding [28].

It can be concluded that the orientation of the PrP domains in the dimers plays a strong role in the determination of their mechanical unfolding behavior, leading to strong changes in the mechanical stability of the folded structures, probably due to the possibility of neighboring C-terminal domains to interact with each other.

## 4. Discussion

In summary, we used AFM SMFS approaches to record a relatively large number of heterogeneous events involving the non-native folding of extended portions of the PrP chain, both when probed in its monomeric and dimeric forms. Our experiments confirmed that the PrP C-terminal domain unfolds with a two-state mechanism without any intermediate, as it was not possible to identify any different pattern of unfolding. The peculiar orientation of the PrP dimers used in this study let us evidence a significant change in the nanomechanical unfolding signals, possibly due to the emergence of stable associative structures involving the otherwise natively folded C-termini.

The relevance of PrP dimers has been presented and argued many times in the literature, often with conflicting results. In early studies, it was suggested that the dimeric form of PrP^Sc^ was the smallest possible size for infectivity [29,30]; conversely, PrP^C^ dimers may exhibit a protecting activity against prion replication [31]. Later, X-ray crystallography studies on the C-terminal PrP domain showed the possibility of dimerization via an interchain disulfide bridge that forms due to domain-swapping, although the physiological relevance of this dimer was not clear [32]. Cell biology studies using engineered constructs expressing covalently linked PrP^C^ propose that PrP^C^ homodimerization might represent a protective dominant-negative mechanism that sequesters PrP^C^ from prion conversion [33,34]. However, the structural events leading to dimer formation, i.e., if the dimer is in an N-C or C-C conformation, are unclear so far. Only recently, two dimeric native PrP forms have been described at the atomic level, called α1 and α3 dimers as a reference to the α-helices involved in the dimerization interface. Notably, dimer formation requires a C-C orientation of each monomer, and this building block might potentially lead to an infinite polymer [35]. The cryo-EM structure of brain derived fibrils supports the model of an amyloid composed of monomeric PrP^Sc^ units disposed in C-C orientation [12].

Recently, Woodside and coworkers studied hamster PrP dimer constructs with DNA handles with OT SMFS [36]. Their findings seem to differ from ours, as they report that dimers lack their native folding. The OT studies are done in different conditions than the AFM and probe proteins in different ranges of force-loading rates. Furthermore, while in our case, the AFM probes each solution-folded molecule only once, the OT experiments commonly perform many unfolding-refolding cycles (on an often more limited number of different molecules) in order to study folding kinetics and build molecule statistics. Woodside and coworkers could describe a number of intermediate misfolded structures in their study thanks to the high force sensitivity of their OT experiments. As in our system it is only possible to probe more mechanically stable structures, pulled at a higher loading rate, our study might be reporting the behavior of the molecular system probed farther away from equilibrium. Our finding of a strongly heterogeneous set of structures involving a large portion of the PrP chain might be the result of this.

Globally, it can be asserted that AFM SMFS data can record a relatively large number of heterogeneous events involving the non-native folding of extended portions of the PrP chain, both when probed in its monomer and in its dimeric forms.

Our work provides the first biophysical description of the unfolding events linked to a dimer form oriented in a C-C conformation. We propose that this early dimeric form could represent a building block of an amyloid fibril. Additional single-molecule experiments and novel constructs harboring more PrP fragments or different internal polyprotein fingerprinting constructs [19] might shed more light on the still obscure initial states of PrP aggregation.

## Figures and Tables

**Figure 1 biology-11-01358-f001:**
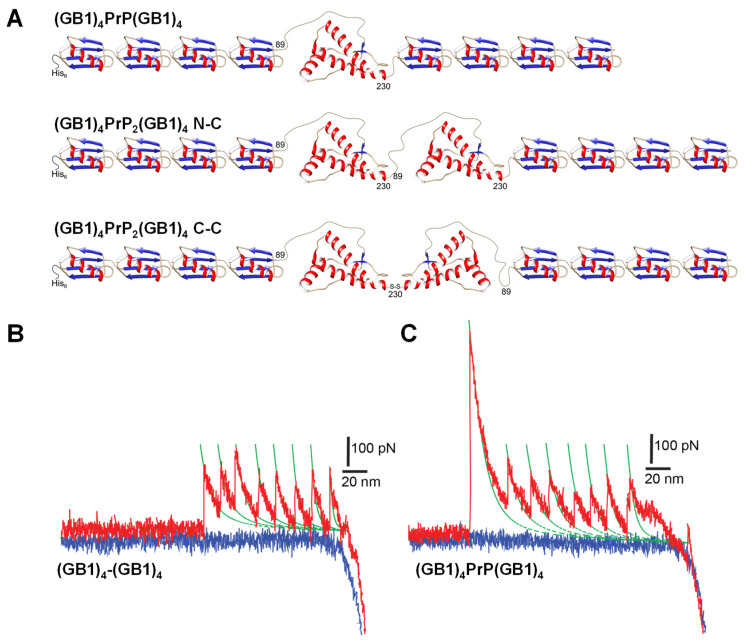
(**A**) Cartoon representation of the main PrP constructs used in this study: monomeric or (GB1)_4_PrP(GB1)_4_ polyprotein, N-C connected dimer or (GB1)_4_PrP_2_(GB1)_4_ N-C, and C-C dimer or (GB1)_4_PrP_2_(GB1)_4_ C-C. (**B**,**C**) Typical force-extension curves (FEC) from unfolding experiments at a neutral pH of (GB1)_4_-(GB1)_4_, i.e., two modules of 4 GB1 separated by a short Arg-Ser linker sequence, and (GB1)_4_PrP(GB1)_4_, respectively. One protein end is picked up during surface contact and stretched until the protein modules are put under tension. The unfolding is registered as an abrupt decrease of force up to the detachment of the protein from the AFM probe. Colored traces in (**B**,**C**) panels represent the recording of the force as a function of the chain extension during piezoelectric stage movements of the AFM: during the approach (blue trace), the cantilever encounters the surface and starts pushing on over it, as indicated by the sudden increase of negative force. Only occasionally the probe tip is then retracted (red trace) with one molecule and, after it stops pushing over then surface, molecules are tethered to it. The molecule is stretched until it finally detaches from the tip (last peak). The green trace represents the worm-like chain fit to each unfolding peak.

**Figure 2 biology-11-01358-f002:**
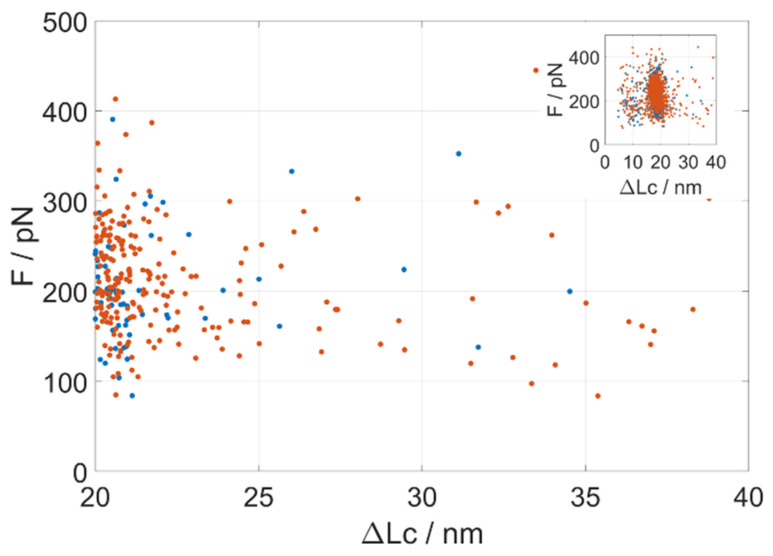
Scatter plot representing two subsets of the recorded force unfolding events for (GB1)_4_-(GB1)_4_. (i.e., the control, blue points) and (GB1)_4_PrP(GB1)_4_ monomeric construct (red points) in the region at unfolding lengths from 20 nm to 40 nm, where GB1 events are scarce, and the main contribution derives from the unfolding of the (GB1)_4_PrP(GB1)_4_ construct. The *x*-axis indicates the contour lengths (ΔLc in nm), i.e., the chain length of folded protein chain that unfolds in the event. The *y*-axis represents the unfolding force (F in pN) applied to unfold the protein module. In the *inset*, the full scatterplot, comprising the dense region where most of the GB1 events occur (see also Appendix A). The kernel density estimation of the events’ probability and their difference are presented in Appendix A.

**Figure 3 biology-11-01358-f003:**
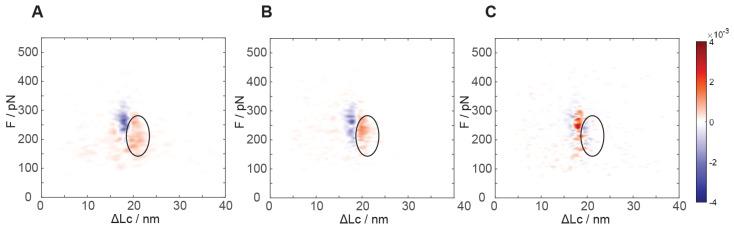
Kernel density estimation (KDE) probability density distribution of the differences in unfolding events frequency between each PrP monomeric sample and the (GB1)_4_−(GB1)_4_ reference computed for the following constructs: sample and (**A**) (GB1)_4_PrP(GB1)_4_ construct, (**B**) (GB1)_4_C_term_−PrP(GB1)_4_ construct, and (**C**) reduced (GB1)_4_PrP(GB1)_4_ construct. The average of the unfolding events per FEC at each length and force was used for the KDE. From a qualitative view-point, positive differences are marked with a darkening red color scale (more events in the construct than in the reference), while negative differences are on a blue scale. The black ellipsoid visually highlights the area of the C-terminal unfolding cluster (19–24 nm and 140–280 pN, defined from the known contour length of PrP and the analysis of the force unfolding events). From a quantitative point of view, the color of the KDE pixels represents (according to the colormap on the side) the interpolated density of probability of finding force-unfolding events in each of the plot pixels (measuring about 0.2 nm by 1 pN). By integrating such density of probability over an area of interest, the normalized (differential) probability of force-unfolding events in that region can be obtained.

**Figure 4 biology-11-01358-f004:**
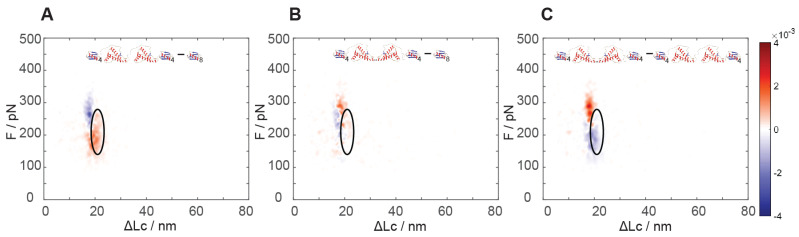
Kernel density estimation of the differences in the normalized probability of unfolding event frequency between (**A**) N-C linked dimeric PrP and (**B**) C-C linked dimeric PrP constructs and the (GB1)_4_−(GB1)_4_ reference sample. The black ellipsoids highlight the C-terminal unfolding region (same as in Figure 3). Unfolding events with a ΔLc > 39.7 nm are related to PrP dimeric associations and are not highlighted here as they occur with a much lower frequency. (**C**) KDE distribution of the differences in unfolding event frequency between the C-C dimeric PrP and the N-C dimeric PrP constructs. In the upper part of the panels, cartoon models of the PrP and GB1 constructs identify the structures for which the event probability differences are estimated. The pixel colors are coded according to the attached colormap representing the density of probability (see also the caption of Figure 3 for further explanation).

## Data Availability

The data that support the findings of this study are available from the corresponding authors upon request.

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
