# Peer review of "Evidence of Orientation-Dependent Early States of Prion Protein Misfolded Structures from Single Molecule Force Spectroscopy"

_biology, 2022, doi:10.3390/biology11091358_

Round 1
Reviewer 1 Report
Summary:
The authors use atomic force microscopy (AFM) (a well-established single-molecule force spectroscopy (SMFS) technique) to probe the mechanical stability and unfolding of recombinant prion protein (PrP) in the monomeric and dimeric state, thereby elucidating the processes governing early oligomerisation. By applying this technique, the authors reveal that: (i) the C-terminal domain of PrP unfolds in a two-state process, (ii) there is considerable heterogeneity in PrP unfolding, (iii) there are differences in unfolding depending on the orientation of the PrP dimer, whereby the C-C dimer unfolds at a higher force compared to the N-C dimer (suggesting higher stability). Thus, the authors propose that the C-C dimer may represent an essential building block of amyloid fibrils.
Understanding how PrP (and other proteins involved in protein misfolding diseases) self-associate is of particular importance, as it may shed light on the initial steps involved in amyloidal aggregation and associated neurotoxicity.
General comments:
Overall, I think the manuscript contains some interesting and potentially useful data. However, the key message of the paper gets a bit lost. I would consider rewording some of the abstract to emphasise the main experimental findings and potential implications of this work. I also recommend that the authors expand their introduction to include a more detailed explanation of AFM-SMFS and why it is a powerful tool for studying protein aggregation in the context of protein misfolding diseases, as a proportion of the readership will not be familiar with these techniques.
In terms of scientific content, I find the data presented in this manuscript to be scientifically sound (with the caveat that I am not overly familiar with the methodologies used). However, certain paragraphs of the results section are difficult to understand and there are some presentation issues that make it hard to interpret the data. For example, I would expect every key point/finding in the results section to refer back to a specific figure, which isn’t always obvious in the manuscript. In addition, some key concepts are not fully explained (e.g., contour length), and a number of figure legends lack sufficient detail to fully describe the accompanying figure.
Specific comments:
Abstract
1. Lines 27-28: Consider changing “atomic force spectroscopy” to “atomic force microscopy” to keep terminology consistent throughout the paper (lines 36, 53, 68, 114, 119, 143, 282, 283 & 292 all refer to “atomic force microscopy/AFM).
2. Lines 30-32: Reword the sentence beginning “Notably, the analysis of two dimeric constructs…”, as the key message of the paper gets lost.
Introduction
3. Lines 40-42: The opening sentence of the Introduction contains a list of neurodegenerative disorders. I recommend the authors add prion diseases to this list, given that PrP is the main subject matter of the paper.
4. Lines 49-52: Add more detail on AFM-SMFS. I suspect that a proportion of the readership will be unfamiliar with these techniques.
5. Lines 52-54: The authors should state which intrinsically disordered proteins (IDPs) they are referring to. Specifically, if some of the proteins involved in the diseases they have mentioned (Alzheimer’s, Parkinson’s, ALS, prion diseases) are IDPs (or proteins that contain intrinsically disordered regions), the authors should make this explicit in the text.
6. Lines 70-72: Define GB1 and expand on the rationale for using GB1 in this study. Is this is a standard approach? If so, make this explicit in the text and add the appropriate references.
Materials and methods
7. Lines 80-83: The authors should make it clear which PrP sequence they are referring to (presumably truncated mouse PrP?). Either include protein sequence(s) in the supplementary methods or add a sequence identifier (e.g., accession number) to the methods section.
8. Lines 90-92: The sentence beginning “The C-C constructs derived…” is unclear. Please reword/clarify.
9. Line 110: I cannot find any circular dichroism data in the results section or supplementary data. If this data is not included in the manuscript, follow the sentence mentioning circular dichroism with (data not shown).
Results
10. Figure 1 (panels B & C): The figure legends lack sufficient detail to fully describe the accompanying figure. The authors should explain what red, blue and green traces/lines refer to in the figure legend.
11. Line 146: Re: “contour lengths”, the authors should explain what they mean by this (don’t automatically assume a reader will know) and briefly mention how contour lengths were calculated/derived. Ideally, this should also be illustrated in Figure 1.
12. Lines 147, 162 & 182: Re: mention of “significant”, “statistically different” or “significant statistical difference”. This suggests that formal statistical analysis has been conducted. If the authors have conducted statistical analysis on certain datasets this should be detailed in the methods section and/or in the appropriate figure/figure legend. If no such analysis has been conducted, these sentences should be reworded.
13. Line 162: Change “C-N dimeric construct” to “N-C dimeric construct” to keep terminology consistent.
14. Fig 2: The figure legends lack sufficient detail to fully describe the accompanying figure. The axes of the scatter plot should be defined in the figure legend.
15. Lines 173-180: This paragraph is confusing. Please reword/clarify.
16. Lines 180-191: The authors refer to “differential analysis of the mechanical unfolding of constructs containing GB1 with or without monomeric PrP…”. However, both of the figures they refer to (Figure 3 & Figure S6) show comparisons between the two dimeric PrP constructs. I recommend the authors clarify this paragraph, including which figures they are referring to (perhaps Figures S2 and S3, which, despite their inclusion in the supplementary materials, are not referred to in the main text of the manuscript).
17. Fig 3: Include a brief explanation of the heat map scale (on the right-hand side of the figure) in the figure legend.
18. Lines 231-234: Re: the sentence beginning “It turned out that about 0.5 events…”. It is not clear how the authors arrived at these figures. Please clarify and (if possible) refer to a specific figure that illustrates this.
19. Lines 244-247: Re: the sentence beginning “An approximation based on the estimated persistence length…”. Again, it is not clear how the authors arrived at these figures. Please clarify and (if possible) refer to a specific figure that illustrates this.
Supplementary materials
20. Figure S5: Change “C-N linked dimers” to “N-C linked dimers” in figure legend to keep terminology consistent.
Reviewer 2 Report
This manuscript describes a detailed characterization of the mechanical unfolding characteristics of Prion protein (89-230) using single molecule force spectroscopy. The constructs studied include a four-copy repeat of a reference artificial elastomeric protein. The authors use an innovative method to change the orientation of the prion protein so that they are able to characterise the dimeric forms of the prion protein in two different orientations. Through this effort, they detected a stable dimeric conformation of PrP when the dimer is oriented in a C-terminal to C-terminal conformation.
The data seem to be very interesting and will provide a much needed platform to perform further studies. I have two questions regarding the authors' experimenal design, which I hope the authors' could shed light upon.
1. Was there any specific reason that the number of GB1 units that the authors chose to link to PrP was fixed at 4? Or was the choice arbitrary? Perhaps the authors might explain their reasons.
2. A check of the UniProt database (https://www.uniprot.org/uniprotkb/P04156/entry) indicates that PrP 89-230 contains a single intramolecular disulfide bond (between 179-214) which would be contained in the fragment that the authors are studying; indeed, the authors discuss the effects of cutting this bond on the unfolding behavior in their Figure S7. My question is, would the presence of this disulfide bond (or lack thereof under certain conditions during purification) complicate the preparation and isolation of the C-C construct? In other words, what is the authors' evidence for concluding that the C-C dimer is indeed forming a disulfide bond between the cysteines introduced at position 231, and not the other cysteines located in the polypeptide sequence? An explanation would be welcome.
